# ASL Citizen: A Community-Sourced Dataset for Advancing Isolated Sign Language Recognition

**Aashaka Desai**[β]    **Lauren Berger**[γ]    **Fyodor O. Minakov**[α]    **Vanessa Milan**[α]
**Chinmay Singh**[α]    **Kriston Pumphrey**[γ]    **Richard E. Ladner**[β]
**Hal Daumé III**[α,δ]   **Alex X. Lu**[α]    **Naomi Caselli**[γ]    **Danielle Bragg**[α]

[α]Microsoft Research  [β]University of Washington  [γ]Boston University  [δ]University of Maryland
{hal3,lualex,dabragg}@microsoft.com
{aashakad,ladner}@cs.washington.edu, nkc@bu.edu

## Abstract

Sign languages are used as a primary language by approximately 70 million D/deaf people world-wide. However, most communication technologies operate in spoken and written languages, creating inequities in access. To help tackle this problem, we release ASL Citizen, the first crowdsourced Isolated Sign Language Recognition (ISLR) dataset, collected with consent and containing 83,399 videos for 2,731 distinct signs filmed by 52 signers in a variety of environments. We propose that this dataset be used for sign language dictionary retrieval for American Sign Language (ASL), where a user demonstrates a sign to their webcam to retrieve matching signs from a dictionary. Through our generalizable baselines, we show that training supervised machine learning classifiers with our dataset achieves competitive performance on metrics relevant for dictionary retrieval, with 63% accuracy and a recall-at-10 of 91%, evaluated entirely on videos of users who are not present in the training or validation sets.

## 1   Introduction

Communication in sign language is an essential part of many people's lives. As 70 million deaf people world-wide primarily use a sign language [57], the meaningful inclusion of signing deaf people requires widespread access to sign languages both in individual communities and society-wide (§2.1). Towards this, over 100,000 students per year enroll in American Sign Language (ASL) classes [42], and the number of UN countries mandating provision of services in sign language has grown from 4 in 1995 to 71 in 2021 [56].

Despite increasing support for equal access to sign language, most existing information resources (like search engines, news sites, or social media) are written, and do not offer equitable access. Requiring signing deaf people to navigate information in a written language like English necessarily means forcing them to operate in a different, and potentially non-native language. Adapting resources for sign language input and output introduces significant technical challenges, which have motivated the development of computational methods for sign language recognition, generation, and translation [36].

In this work, we focus on ASL dictionary retrieval. Many existing ASL dictionaries catalog signs with English glosses: an out-of-context translation of a sign into one or more English words (e.g., LOBSTER, CLIMB_LADDER or HAD_ENOUGH). However, English lookup relies on knowing the English translation, and on a 1:1 relationship with English words, which often does not exist. Demonstrating a sign to look it up may be more natural, but is computationally more challenging due to the rich visual format and linguistic complexities. We seek to help address this problem of video-based dictionary retrieval, where a person demonstrates a single sign by video, and the system

returns a ranked list of similar signs. Dictionary retrieval fills a need for language learners, who may see a sign but not know the meaning. Moreover, dictionaries can contribute to documenting sign languages, and allow established signers to navigate dictionary resources directly in sign language.

To help advance dictionary retrieval, we present a crowdsourced dataset of isolated ASL signs, to support data-driven machine learning methods overcome limitations of prior isolated sign language recognition (ISLR) datasets (see Table 1 and §2.2). Model development typically requires a large, high-quality training set (i.e. large vocabulary, minimal label noise, representation of diverse signers and environments). Existing datasets are often filmed in a lab or scraped from online sources, which limit scale and diversity. Web scraping often does not acquire participant consent, which erodes community trust and leads to labeling challenges. To overcome such limitations, we build upon recent crowdsourcing proposals [10] to collect and release *ASL Citizen* – the first large-scale crowdsourced sign language dataset. Our dataset is the largest isolated sign dataset to date, newly representative of real-world settings and signer diversity, and collected with permission and transparency.

Using this new dataset, we adapt previous approaches to ISLR [41, 51] (§2.3) to the dictionary retrieval task, and release a set of baselines for machine learning researchers to build upon (§4). Dictionary retrieval requires algorithms to return a ranked list of signs, given an input video. A variety of methods can satisfy this output, but we focus on supervised deep learning methods, taking advantage of recent ISLR methods. We show that even without algorithmic advances, training and testing on our dataset doubles ISLR accuracy compared to prior work, despite spanning a larger vocabulary and testing on completely unseen users (§5). We also evaluate our dataset against prior datasets by comparing performance on a subset of overlapping glosses, and by comparing performance of learned feature representations from models trained on these datasets, showing further improvements in each case. Finally, through a series of downsampled training set experiments, we show that dataset size contributes to our improved performance.

Throughout this research, we take a culturally-sensitive and participatory approach to sign language computation. Sign languages are a cornerstone of Deaf culture and identity.[1] Previous works have noted that many efforts in sign language computation promote misconceptions about sign language, exploit sign language as a commodity, and undermine Deaf political movements seeking recognition of sign languages [62, 7, 21, 5]. Other works question if technologies being designed actually benefit Deaf communities, and document patterns where new tech is rejected by signing communities for being intrusive, clunky, or insensitive [29, 39]. This work abides by calls issued by disability scholars for better collaboration with Deaf communities, and for focus on solving real needs [29, 7, 62, 9], and demonstrates that aligning with these calls can actually help advance machine learning results.

In summary, our primary contributions are:

1. We provide a benchmark dataset and metrics for the dictionary retrieval task, with released code. Not only does this application have real utility to the signing community, but it grounds ISLR in a real-world problem setting, informing data collection and metrics.

2. We release the first crowdsourced dataset of isolated sign videos, containing diverse signers in real-world settings representative of real-world dictionary queries. It is about four times larger than prior isolated sign datasets, and the only of its scale collected with Deaf community involvement, consent, and compensation.

3. Our general baselines improve over ISLR accuracy achieved by prior datasets by more than double while using the same architectures, highlighting the impact of appropriately collected data on model performance.

For links to the dataset, code, and additional supplementary materials please visit https://www.microsoft.com/en-us/research/project/asl-citizen/.

## 2 Background and Related Work

### 2.1 Sign Languages and Deaf Culture

Sign languages are complex, with large vocabularies and unique phonological rules. Analogous to the sounds of speech, signs are composed of largely discrete elements (e.g., handshape, location, and

---

[1]By established standards, we use "Deaf" to refer to cultural identity, and "deaf" to audiological status.

| Dataset | Vocab size | Videos | Videos/sign | Signers | Collection | Consent |
|---------|-----------:|-------:|------------:|---------|------------|:-------:|
| RWTH BOSTON-50 | 50 | 483 | 9.7 | 3 Deaf | Lab | ✓ |
| Purdue RVL-SLL | 39 | 546 | 14.0 | 14 Deaf | Lab | ✓ |
| Boston ASLLVD | 2,742 | 9,794 | 3.6 | 6 Deaf | Lab | ✓ |
| WLASL-2000 | 2,000 | 21,083 | 10.5 | 119 Unknown | Scraped | ✗ |
| **ASL Citizen** | **2,731** | **83,399** | **30.5** | **52 Deaf/HH** | **Crowd** | ✓ |

Table 1: Prior ISLR datasets for ASL compared to ASL Citizen. HH stands for hard of hearing.

movement) according complex rules [12]. English translations of isolated signs are called glosses, are written in all-caps, and may be single words or multiple words, typically not mapping 1-1 to English (just as in any other language translation). Sign execution varies across contexts, signers, and sociolinguistic groups [44]. These factors complicate representative data collection and modeling.

Isolated signs like those in dictionaries are considered "core" parts of the lexicon, but are only a subset of sign languages [11]. The "non-core" lexicon is generally not well represented in dictionaries or lexical databases, and includes complex constructions like depicting verbs, classifier constructions, and verbs that use time and space in ways that can be difficult to decompose into discrete parts [65, 25]. Continuous signing also includes coarticulation and grammar expressed with the face, body, and signing space, in addition to the hands [59]. As such, ISLR only address a fraction of sign language recognition. However, since our goal is dictionary retrieval, this work focuses on isolated signs.

Sign languages play a critical cultural role in Deaf communities and identity [27]. While our work focuses on American Sign Language (ASL), which is primarily used in North America, over 300 sign languages are used worldwide. Sign languages have been suppressed by political and educational authorities to force deaf individuals to integrate into hearing society by favoring speech at the expense of individual welfare [40]. These oppressive movements promote misconceptions that persist today (e.g., that sign languages are lesser languages, or that ASL is signed English), and Deaf activists work to combat these ideas [29, 21, 62]. This cultural context informs our decision to formulate ISLR as a dictionary retrieval problem, which grounds research in a meaningful real-world use case.

## 2.2 Previous ISLR Datasets

Our work focuses on ASL, which has four main public ISLR datasets: WLASL [41], Purdue RVL-SLL[58], BOSTON-ASLLVD [3] and RWTH BOSTON-50 [63], summarized in Table 1. WLASL offers four different vocabulary sizes, the largest containing 2,000 signs (WLASL-2000 in our tables). While BOSTON-ASLLVD contains a larger vocabulary of 2,742 signs, the number of videos per sign is limited. Real-world signs vary greatly by user due to dialectal (e.g., geographic region) and sociolectal (e.g., age, gender, identity) variation. Models trained on a small number of dataset contributors, as seen in prior work, may not generalize well to diverse signers [3, 63, 58].

Existing datasets use varied collection and labelling techniques, impacting quality and size. Lab-collected data [58, 63, 3] is usually high-quality with clean labels, but limited in size, participant diversity, and real-world settings. Data scraped from the internet may capture more users or settings, but varied contributor fluency and difficulty identifying and segmenting signing in videos impacts quality. Including ASL teaching resources and relying on English text therein (e.g. [41, 37]) still creates unreliable labels, as even linguists struggle to label signs without a conventional notation system [24, 32]. Teaching videos are also often professionally-recorded, with similar limitations to lab data. Moreover, scraping is typically not allowed by content creators or hosting platforms.

Sign language dataset design has profound implications for responsible AI [9]. Videos feature identifiable faces and are expensive and labor-intensive to create; contributor consent is paramount. Prior work has proposed crowdsourcing [10] and addressing privacy concerns [8] as ways to collect larger representative datasets. Our work explores such ideas at scale, implementing a crowdsourcing platform with optimized versions of tasks proposed in [10] and partnering with Deaf community throughout. As a result, we present the first crowdsourced sign video dataset containing a large vocabulary and representative signers in everyday settings, collected with consent.

## 2.3 Isolated Sign Language Recognition Methods

Research on isolated sign language recognition (ISLR) is increasing, as evinced by the growing number of literature reviews in the space [36, 17, 55, 18, 53, 22, 48, 38, 1]. Early approaches relied on handcrafted features and classic machine learning classifiers [54, 26, 46, 14, 45], typically on small datasets and vocabularies. More recent approaches have shifted to deep learning as larger datasets have become available. Appearance-based methods operate directly on video frame inputs: approaches include spatially pooled convolutional neural networks [47, 41] and transformers [20, 4]. Alternatively, pose-based methods [41, 51, 4] fit models on keypoints extracted using human pose models (e.g. OpenPose [13], MediaPipe [43]). Despite the breadth of approaches, state-of-the-art ISLR methods still have relatively low recognition performance, Prior baselines for ISLR using standard approaches achieve about 30% accuracy [41, 51] over realistic vocabulary sizes (2,000+ signs), which is largely inadequate for real-world use. More recent works have improved performance with algorithmic advancement: for example, using multiple modalities simultaneously (optical flow, depth map, depth flow) [35], or by introducing new priors (such as hand modeling) [33], or by leveraging cross-lingual semantic information to learn relations between classes[64]. While these models achieve better performance (about 60% accuracy), the complexity of these methods may obscure the impact of the dataset. Thus for our baselines, we use more generalizable techniques as seen in prior dataset works (such as I3D and ST-GCN) [15, 61].

## 2.4 Sign Language Dictionaries

Dictionaries are a meaningful application of ISLR to Deaf community members [34, 7]; they play a cultural role in language documentation, and are valuable tools for language users and learners. However, creating effective sign lookup systems is difficult. English-to-ASL dictionaries (e.g. [52]) accept written queries and so can leverage text-based matching methods, but ASL-to-English and ASL-to-ASL dictionaries cannot because no standard sign language writing system exists. This then leaves sign language users to operate in a non-native language to access dictionary resources. Similarly, novice signers may not know the English gloss to leverage English-to-ASL dictionaries.

To address these challenges, two main approaches are used in ASL-to-English dictionaries: feature-based lookup and example-based lookup. In feature-based lookup, users specify parameters of the sign they seek (e.g., the handshape, the body location involved, the motion, etc.), and a list of top matching signs are returned [6]. While this simplifies the lookup problem, unlike English spelling, these features are not conventional and are not widely used or taught, so users may not be familiar with how to make use of them. Novice learners may especially have trouble noticing and remembering these parameters.

In example-based lookup, users demonstrate a sign by video, and receive a list of top matching signs. While this may be more accessible for users, it is more challenging computationally, now requiring video processing to complete lookup [60]. Because this is largely unsolved, human-computer interaction (HCI) research has primarily focused on understanding potential dictionary use through wizard-of-oz methods [31, 2, 30, 28]. Our work advances ISLR for dictionary retrieval using generalizable methods, potentially enabling functional example-based dictionaries to be created and studied.

# 3 ASL Citizen Dataset Creation

## 3.1 Data Collection

We build on prior work [10], which piloted the first crowdsourcing tasks for sign language data collection in a small user study. In this work, we scale data collection with a longer-term deployment with fluent Deaf signers. We also enhanced the crowdsourcing tasks proposed in [10] to improve recording efficiency and data quality (details in Appendix). This method secures participant consent, and curates data appropriate for work on dictionary lookup; participants are asked to contribute videos for a communal dictionary, recording videos in real-world settings, similar to real-world dictionary queries. The task design also eliminates labelling challenges by collecting pre-labelled content.

The platform provided each user with the full vocabulary of 2,731 signs (from ASL-LEX [16, 50]), sorted so that signs with the fewest videos are first, to help balance the dataset across signs. For each

|  | Train | Val | Test |
|---|---|---|---|
| Users | 35 | 6 | 11 |
| Videos | 40,154 | 10,304 | 32,941 |
| User distrib. | 60% F | 83% F | 55% F |
| Video distrib. | 54% F | 71% F | 55% F |

Table 2: Statistics for ASL Citizen dataset splits.

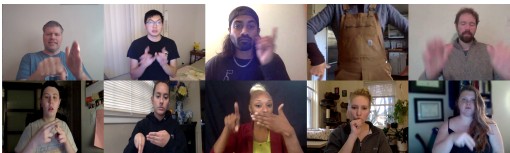

Figure 1: Random ASL Citizen video stillframes.

sign, they first viewed a "seed" video of the sign filmed by a highly proficient, trained ASL model. This "seed signer" was a paid research member, and used a high-resolution camera. Participants then recorded their own version of the demonstrated sign. As in [10], users could re-play videos, and re-record or delete. We provided optional English gloss, hidden by default to encourage focus on the ASL. For every 300 videos, participants received a $30 gift card, for up to 3,000 signs. Those who completed the vocabulary re-visited the least-recorded signs. Providing demographics was optional.

We took steps to help ensure that our data collection was culturally sensitive and participatory. Deaf researchers were involved in every aspect of the research, and made direct contact with participants. All recruitment and consent materials were presented in an ASL-first format, featuring short ASL videos. Participants were recruited through relevant email lists and snowball sampling of Deaf researchers' social networks. All procedures were reviewed and approved by IRB.[2]

## 3.2 Data Verification and Cleaning

To help ensure data quality, we engaged in verification and cleaning procedures under close guidance from our ethics review board. First, we removed empty videos automatically, by removing those under 150 KB in size or where YOLOv3 [49] did not detect a person ( 50 videos). We manually reviewed the first and last videos recorded by each participant on each day and random samples throughout, checking for a list of sensitive content provided by our ethics and compliance board. Three types of personal content were identified for redaction: another person, certificates, and religious symbols. To protect third parties, we used YOLOv3 to detect if multiple people were present. For these videos and others with identified personal content, we blurred the background using MediaPipe holistic user segmentation. We blurred a subset of pixels for one user, since the personal content was reliably limited to a small area. We also removed one user's videos, who recorded many videos without a sign in them, and videos of a written error message. Finally, we manually re-reviewed all videos.

In our reviews, we did not identify any inappropriate content or bad-faith efforts. In total, we blurred the background of 268 videos where a second person was detected automatically, 293 additional videos with sensitive content, and 32 additional videos with a person missed by the automatic detection. We blurred a small fixed range of pixels for 2,933 videos, and omitted 513 videos where the blurring was insufficient or an error message (resulting from the data collection platform) showed.

## 3.3 ASL Citizen Dataset Benchmark for Dictionary Retrieval

Our final cleaned dataset, ASL Citizen, contains 83,399 videos corresponding to 2,731 signs recorded by 52 participants, including the seed signer (Table 1).[3] This is the first crowdsourced ISLR dataset, and the largest for ASL (or any other sign language) to date. Our vocabulary was taken from ASL-LEX [16], a database of lexical and phonological properties of signs. As such, each ASL Citizen video maps to an ASL-LEX entry, enabling the use of ASL-LEX's linguistic annotations (e.g., the handshape of each sign in our dataset). Each sign has multiple recordings ($\mu = 30.5, \sigma = 1.7$).

Of our participants, 49 identify as Deaf and 3 as hard of hearing; 32 as female and 20 as male; and with ages 20-72 years old ($\mu = 36.16, \sigma = 14.2, n = 49$). These signers come from 16 U.S. states, with 2-65 years of ASL experience ($\mu = 30, \sigma = 15.12, n = 48$), with mean ASL level of 6.45.

We expect our videos to be consistent with our chosen dictionary retrieval task (i.e. participants resemble webcam users demonstrating signs to a dictionary). First, all users were informed they were contributing to a dictionary. Second, participants recorded videos in a variety of everyday settings,

---

[2]Microsoft and Boston University IRBs reviewed the project. Microsoft serves as the IRB of record (#418).
[3]In this paper, we use ASL Citizen Version 1.0. All stats and experiments were run with this dataset version.

spanning varied illumination, background, resolution, and angle. Since our videos are self-recorded, there is also variability in when users start and finish signing in videos (i.e. amount of padding), and the speed, repetition, and execution of signs. We consider this variability to be valuable as it is within the scope of "in-the-wild" dictionary queries, so we do not filter or standardize these variables.

**Dataset Splits.** We also release standardized splits of our 52 users into training, validation and test sets (Table 2), attempting to balance by female-to-male gender ratio (no participants self-identified as non-binary). Importantly, we establish splits such that users in the validation and test sets are unseen during training. While previous work randomly split videos [41] (so users in the test set may be seen in training), we felt it critical to evaluate on unseen users to align with our dictionary retrieval problem; it is unlikely that a user looking up a sign would be in the training set of a deployed model. Accordingly, the seed signer is placed in the training split. We also sought to provide a large test set. While still leaving sufficient videos for training (over $2.5\times$ that of the previous largest dataset [41]), a large test set with multiple users not only offers more robust performance estimates, but also offers the future potential for a wider range of methods (e.g., unsupervised domain adaptation).

**Sign Ranking and Metrics.** Dictionary retrieval requires models to return a ranked list of signs, so we consider standard information retrieval metrics: recall-at-K (for K=1, 5, and 10, where recall-at-1 is the same as accuracy), discounted cumulative gain (DCG) and mean reciprocal rank (MRR). Recall-at-K, measured by determining if the correct sign is in the top K rankings, allows us to consider scenarios where users may look at only the first K signs returned for their query [30]. Alternatively, DCG and MRR evaluate the overall ranking of the correct sign in the entire list. For all these metrics, a higher score indicates the correct sign is earlier in the ranking. DCG uses a log scale, so is more sensitive to order at the top of the ranking. The Appendix provides formulas and additional details.

# 4 Methods and Training Details

We train two fundamentally different types of machine learning methods on the ASL Citizen dataset: an appearance-based approach, I3D, which is based on a 3D convolutional network applied directly to the video frames [15]; and a pose-based approach, ST-GCN, which preprocesses the video to extract pose information, on which we train a temporal graph convolutional network [61]. By choosing simple representative models, I3D and ST-GCN, we aim to showcase the quality of the data while minimizing the assumptions encoded into the methods. Both are classifiers with an output space equal to the size of the vocabulary. For both, to generate a ranked list of retrieved signs, we sort the output probabilities across labels.

**Data Preprocessing** For our I3D model, we preprocess videos by standardizing to 64 frames. Due to variance in sign length and user execution, our dataset videos differ in length. While previous works use random temporal crops to standardize frame lengths during training [41], we reasoned this practice might alter sign semantics: some signs are compounds of multiple signs, and temporal cropping may reduce the sign to just one root sign. Instead, we standardize training and evaluation videos by skipping frames: for videos longer than 95 frames, we skip every other frame, and for videos longer than 159 frames, we take every third frame. Videos shorter than 64 frames are padded with the first or last frame, and longer videos had even numbers of frames removed from the start and end. Finally, we augmented with random horizontal flips to simulate left and right handed signers.

For our ST-GCN model, we extracted keypoints using MediaPipe holistic. We use a sparse set of 27 keypoints previously established by OpenHands [51]. Following previous practice, extracted keypoints are center scaled and normalized using the distance between the shoulder keypoints. Since our videos contain a higher frame-rate than ISLR datasets analyzed by OpenHands, we cap the maximum frames to 128, and downsample frames evenly if the video is longer. As with OpenHands, we apply random shearing and rotation transformations during training as data augmentation.

**Model Structure and Training** We train our I3D model for up to 75 epochs using learning rate 1e-3 and weight decay 1e-8, with an Adam optimizer and ReduceLRonPlateau scheduler with patience 5. As we observed that a cross-entropy loss on the entire video alone led to poor convergence, we also employ a weakly supervised per-frame loss previously used in the Charades Challenge [19], where the cross-entropy loss is also applied to each video frame. Our final loss is an average of the full-video cross-entropy loss and this per-frame cross-entropy loss. We train our ST-GCN model

| Model | Train Data | Test Data | DCG | MRR | Rec@1 | Rec@5 | Rec@10 |
|-------|-----------|-----------|-----|-----|-------|-------|--------|
| I3D | ASL Citizen | ASL Citizen | **0.7913** | **0.7332** | **0.6310** | **0.8609** | **0.9086** |
| ST-GCN | ASL Citizen | ASL Citizen | 0.7637 | 0.6997 | 0.5952 | 0.8268 | 0.8813 |

Table 3: Appearance and pose-based baseline results, representing a phase transition in ISLR, up from mid-30% accuracy (Rec@1) in prior work. The best results on ASL Citizen test set are **bold**.

for a maximum of 75 epochs using a learning rate of 1e-3 using an Adam optimizer and a Cosine Annealing scheduler. For each model, we selected the best-performing checkpoint on the validation set for analysis on our test set. Code and model weights are released publicly alongside our dataset.

## 5    Results and Analysis

### 5.1    Novel Benchmarks

We trained our appearance-based I3D model [15] on ASL Citizen to establish an initial baseline, with 63.10% top-1 accuracy (recall-at-1), DCG 0.791, and MRR 0.733 (first row in Table 3). This accuracy is notable, given our problem difficulty – our dataset has completely unseen users, and spans one of the largest vocabulary sizes in ISLR to date (2,731 signs). The pose-based ST-GCN model performs similarly, but consistently worse by a few percentages across metrics, but still substantially better than any previous reported results on datasets of similar size and complexity. For comparison, due to the number of classes, random guessing would yield 0.04% expected accuracy.

In previous work, appearance-based and pose-based models have generally shown competitive performance. Pose-based methods reduce information and potentially introduce errors during keypoint extraction, at the benefit of making relevant features more accessible and standardized compared to raw pixels. However, this standardization may not outweigh errors and information loss when the training set is diverse enough to train general appearance-based methods. Consistent with this, our pose-based model performance slightly lags behind that of our appearance-based model.

### 5.2    Comparison to Prior Datasets

We subsequently seek to understand how much our dataset advances the performance of ISLR models in ASL, compared to previous datasets. To do this, we compared our model trained on our dataset, to a public model trained on the previously largest ASL ISLR dataset, WLASL-2000 [41]; see Table 1. We made no changes to model architecture compared to the WLASL I3D model, meaning that any improvements are because of the training data, not model capacity.

Directly comparing these models is challenging because they use independent gloss mappings. As a result, the number of classes differs, and the same English gloss in one model may refer to a different sign in the other. We overcome this challenge by comparing the models in two ways (Table 4).

First, we directly compared metrics on our test set to previously reported accuracy on the WLASL-2000 test set. While this comparison does not account for potential differences in test set difficulty, we believe our test dataset is more challenging than that of WLASL-2000. Unlike WLASL-2000, we evaluate on unseen signers, and our vocabulary is larger. Comparing top-1 accuracy, we achieve 63.10% on the ASL Citizen test set, while [41] reports 32.48% on the WLASL-2000 test set.

Second, we reduced our test set to a subset of glosses that refer to the same sign in ASL Citizen and WLASL-2000, and used this test set to compare models trained on ASL Citizen and WLASL-2000. We used a reduced version of ASL Citizen's test set to ensure it would not contain anyone seen in training by *either* model, enabling a more fair comparison. We excluded any sign with a documented variant in either dataset, and looked for exact gloss matches across datasets. This produced a set of 1,075 glosses ("Subset" in our tables). To verify matches, an author fluent in ASL examined one example per sign from each dataset for 100 random signs, and did not find any discrepancies. To control for models outputting different numbers of classes, we recalculated the softmax on only the logits for the 1,075 overlapping glosses (effectively excluding predictions outside this set). Under this comparison, top-1 accuracy is 74.16% for our model vs. 8.49% for the WLASL-2000 model. We hypothesize that the accuracy drop for WLASL-2000, also relative to its original test set (32.48%), is because the model was originally evaluated on seen users, and fails to generalize to unseen users.

| Model | Train Data | Test Data | DCG | MRR | Rec@1 | Rec@5 | Rec@10 |
|---|---|---|---|---|---|---|---|
| I3D | ASL Citizen | ASL Citizen | **0.7913** | **0.7332** | **0.6310** | **0.8609** | **0.9086** |
| I3D | WLASL-2000 | WLASL-2000 | -- | -- | 0.3248 | 0.5731 | 0.6631 |
| I3D | ASL Citizen | Subset | *0.8631* | *0.8228* | *0.7416* | *0.9252* | *0.9529* |
| I3D | WLASL-2000 | Subset | 0.2794 | 0.1477 | 0.0849 | 0.2011 | 0.2744 |
| ST-GCN | ASL Citizen | ASL Citizen | 0.7637 | 0.6997 | 0.5952 | 0.8268 | 0.8813 |
| ST-GCN | WLASL-2000 | WLASL-2000 | – | – | 0.2140 | – | – |
| I3D Features | ASL Citizen | ASL Citizen | 0.7652 | 0.6985 | 0.5803 | 0.8450 | 0.9023 |
| I3D Features | WLASL-2000 | ASL Citizen | 0.3153 | 0.1791 | 0.0985 | 0.2528 | 0.3433 |
| I3D Features | Kinetics | ASL Citizen | 0.1234 | 0.0128 | 0.0041 | 0.0136 | 0.0227 |
| I3D Features | ASL Citizen | Subset | 0.8494 | 0.8039 | 0.7107 | 0.9216 | 0.9565 |
| I3D Features | WLASL-2000 | Subset | 0.4039 | 0.2705 | 0.1649 | 0.3785 | 0.4852 |

Table 4: Results comparing across datasets and feature representations: ASL Citizen, prior WLASL-2000 datasets, and a Subset of ASL Citizen "matched" to WLASL-2000. Encoding of best results for test sets - ASL Citizen: **bold**, Subset: *italics*, other datasets: underlined. Gray rows are from Table 3.

| | DCG | MRR | R@1 | R@5 | R@10 |
|---|---|---|---|---|---|
| 0% | 0.103 | 0.002 | 0.000 | 0.002 | 0.004 |
| 25% | 0.393 | 0.262 | 0.162 | 0.365 | 0.467 |
| 50% | 0.679 | 0.597 | 0.481 | 0.736 | 0.809 |
| 75% | 0.763 | 0.698 | 0.592 | 0.830 | 0.880 |
| 100% | 0.791 | 0.733 | 0.631 | 0.861 | 0.907 |

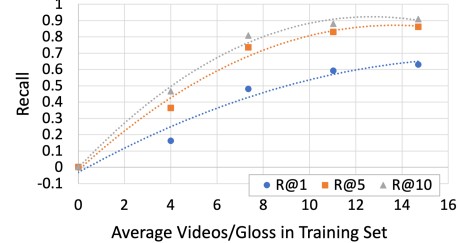

Table 5: Downsampling results. More videos per sign improves performance.

Figure 2: Impact of training data size on recall. We fit a polynomial function to each series.

Together, these results showcase the quality of data in ASL Citizen. Even with standard ISLR algorithms like I3D and ST-GCN, we achieve over 63% top-1 accuracy. In the most naive comparison with independent test sets, this indicates a significant improvement (approx. 2x) in performance compared to prior datasets (63.10% vs. 32.48%). In more standardized comparisons where both models are evaluated on the same test set, our model outperforms prior results by 8.7 times (74.16% vs. 8.49%).

### 5.3 Comparison of Learned Feature Representation

We reasoned that while the classifiers of the ASL Citizen and WLASL-2000 models may predict different sets of signs, both models may still be learning features to recognize signs more generally. To assess this, we extracted the globally pooled feature representation before the classification layer of each model, and built a nearest-neighbor classifier using the cosine distance from ASL Citizen training set representation. We report accuracies on both the full ASL Citizen test dataset, and the reduced overlapping version (Table 4). For the full test set, our model achieves 58.03% top-1 accuracy, while the WLASL-2000 model achieves 9.85%. For the reduced overlap, our model achieves 71.07% top-1 accuracy, while the WLASL-2000 model achieves 16.49%. This suggests that our dataset enables learning a more robust representation of ASL. We similarly test the performance of I3D feature representations trained on Kinetics (an action recognition dataset) [15]: the reported results are close to random, evincing that out-of-domain models do not generalize to sign language recognition.

### 5.4 Impact of Dataset Size

Finally, to understand how scaling data collection affects model performance, we systematically downsampled the training dataset. We generated three random splits: 25% (10,924 videos), 50% (20,077 videos), 75% (30,116 videos) of the original training dataset. We ensured that each sign was represented by at least 4 samples, and smaller splits were subsets of larger splits but otherwise sampled at random. We trained an appearance-based I3D model on each of these splits and report results in Table 5, and also experimented with a 0-shot model with no training data by testing performance on

an I3D model with randomly initialized weights. Our results confirm that the scale of our training set is critical to performance; as training set size decreases, so does accuracy on our test set.

## 6 Discussion and Conclusion

In this work, we introduce a problem formulation for ISLR in the form of dictionary retrieval, provide the largest crowdsourced ISLR dataset to date, and release metrics and baselines showing that our new dataset achieves competitive performance for ISLR in ASL.

We caution dataset users against using ASL Citizen to solve sign language translation. Unlike ISLR, continuous sign recognition requires a system to handle co-articulation, fingerspelling, facial expressions, depictions, and classifiers constructions. Translation requires not only continuous sign recognition, but also the ability to move between ASL and English grammar. Researchers without domain expertise may assume that tokenizing a video into a sequence of signs and applying ISLR to these tokens is sufficient for translation. Underestimating sign language complexity in translation work has led to objections from Deaf communities [29, 39, 21, 23]. The dataset presented here enables technologies like dictionary search, that are focused on classifying signs and do not require considering syntax or even optimal English translation. For example, beyond dictionaries, ISLR can be used to create ASL-first user interfaces that enable a user to interact with signed commands or short responses. We discourage researchers from using this dataset alone (e.g., without also learning from continuous datasets) for more complex applications.

Our data collection process reflects the dictionary lookup problem formulation and yields a large-scale, high-quality ISLR dataset. Traditional lab collections offer high-quality data but limit diversity, and scraped data may promise diversity while introducing labelling and fluency errors and compromising consent. Our crowdsourcing method overcomes such limitations. First, our web platform allows users to contribute from everyday spaces, capturing real-world diversity and use of signing space representative of dictionary applications. Second, by prompting contributors with specific signs to demonstrate, we can generate automatic labels with high confidence. Third, recruiting fluent signers from trusted groups helps ensure quality and capture of ASL conventions. Our study informs future data collection efforts: participatory approaches with meaningful contribution from Deaf researchers can yield not just larger datasets, but higher-quality data.

### 6.1 Limitations and Future Work

While our benchmark models achieve high levels of ISLR performance on unseen signers, future work is still required to fully solve the real-life dictionary retrieval problem. In particular, real-life dictionaries present many use cases not fully addressed in this work.

First, our supervised models operate on a fixed vocabulary, and our benchmarks are unable to cope with signs outside of this vocabulary. Since sign languages are dynamic, with new signs emerging regularly, future work should consider methods that can adapt to signs that were unseen at training time. Second, we evaluated our models on fluent Deaf signers who can expertly replicate signs. Novice signers would likely have difficulty recalling and executing a sign. Therefore, we expect a performance gap for these users, which future models could address. As the vocabulary of ASL is in the magnitude of thousands, further data collection efforts can help create dictionaries more representative of the full vocabulary. While we capture a variety of users and environment in our videos, future work also includes collecting a diversity of signing styles and capturing variations that stem from geographical and learning background to support all users.

Future work also includes deepening evaluation. While we considered DCG, MRR, and recall-at-K in this work, these metrics may not fully align with user preferences. Metrics that better capture overall list relevance (e.g., by weighting relevance of signs with similar meanings or visual appearance high in the list) may reflect a better user experience [30]. Deployed dictionaries also must meet performance measures beyond accuracy (like ease of use or speed). Finally, although we report overall performance, our I3D model accuracy ranges substantially (43-75%) across users. Real-world applications are becoming increasingly viable, and future work should explore whether ISLR models are *equitable*, and how performance disparities between demographic groups might be addressed.

## Acknowledgments and Disclosure of Funding

First and foremost, we thank the community members who participated in this research project by recording themselves signing. We also thank Mary Bellard, Miriam Goldberg, Hannah Goldblatt, Paul Oka, Philip Rosenfield, Bill Thies, and the Microsoft Research outreach team for thoughtful discussions, support, and contributions to the crowdsourcing platform.

This work was supported in part by the National Science Foundation Grants: BCS-1625954 and BCS-1918556 to Karen Emmorey and Zed Sehyr, BCS-1918252 and BCS-1625793 to Naomi Caselli, and BCS-1625761 and BCS-1918261 to Ariel Cohen-Goldberg. Additional funding was from the National Institutes of Health National Institute on Deafness and Other Communication Disorders of and Office of Behavioral and Social Science Research under Award Number 1R01DC018279.

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

# A   Platform modifications

Changes to the platform introduced in prior work [8], with the goal of helping to meet our data collection needs are described below.

| Platform change | Intended impact on data collection |
|---|---|
| Improved recording task flow (automatic countdown to recording start, video upload in the background) | Scale: a more efficient contribution process enables participants to contribute more videos in a set amount of time |
| Visual design overhaul (updated color scheme, button design, and page layouts; replaced one-off video players with standard video players) | Scale: a better user experience attracts and encourages use, and increases trust in the platform creators, which lowers barriers to contributing |
| Updated platform structure and navigation (updated landing page with direct links to contribute; updated navigation links in header; updated info page with more project details in ASL and English) | Scale: a reduced learning curve lets participants contribute more data, and increased trust lowers barriers to contributing |
| Infrastructure scaling (set up infrastructure to handle many parallel contributions, large data storage, backup scripts) | Scale: provides technical capabilities to collect data at scale |
| Removed participant ability to share content with one another in real-time (instead creating two-phased implementation of collection followed by release) | Scale: removes potential for viewing off-putting content from other users |
| New set of seed sign videos executed by a well-known fluent signer who is not white-presenting | Participant diversity: representation creates a more welcoming environment and fosters contributions from a wider range of participants |
| Improved personal data view (enabling users to search through their own videos, sorting videos and demographics into tabs) | Data quality: participants can more easily review and update contributions |
| Overlaid the outline of a human figure on the webcam feed | Data quality: videos are more likely to capture the entire upper body and are more standardized across participants |

Table 6: Summary of platform feature changes, alongside potential impacts on the contributor and resulting dataset.

# B   Dictionary Retrieval Metrics

For a given query, if *i* is the placement of the desired gloss in the returned list of glosses, we calculate metrics using the following formulae:

- Discounted Cumulative Gain = $\frac{1}{\log_2(i+1)}$, ranges in $[\epsilon, 1]$ with 1 indicating that the correct gloss is always the top ranked item, and $\epsilon = \frac{1}{\log_2(N+1)}$ is the smallest attainable score when the correct gloss is ranked last (in our case, $N = 2729$ so $\epsilon = 0.088$). A completely random ordering with give an average DCG of approximately $0.15$.

- Mean Reciprocal Rank = $\frac{1}{i}$, ranges in $[\epsilon, 1]$ with 1 indicating that the correct class is always the top ranked item, and $\epsilon = \frac{1}{N} = 0.00037$. A completely random ordering will give an average MRR of approximately $0.0032$.

The overall scores reported are averages of these metrics across respective data splits (e.g., average over all test instances).

