# OpenReview forum: "ASL Citizen: A Community-Sourced Dataset for Advancing Isolated Sign Language Recognition"
_NeurIPS.cc/2023/Track/Datasets_and_Benchmarks — NeurIPS 2023 Datasets and Benchmarks Poster_

### Official Review · Reviewer_RCuQ · 2023-07-20
**Comments for paper 923**

**Rating:** 6
**Confidence:** 3
**Correctness:** The reviewer thinks that this dataset…
**Clarity:** This paper is easy to follow.

**Strengths:**

The proposed dataset makes ISLR take a further step to the real-world problem setting.

This work releases the first crowdsourced dataset of isolated sign videos.

This work focuses on video-based dictionary retrieval, where a person demonstrates a single sign by video, and the system returns a ranked list of similar signs.

The proposed dataset and dictionary retrieval strategy can effectively improve the state-of-the-art ISLR accuracy.

**Additional Feedback:**

N/A

**Documentation:**

The details are enough and the URL is available.

**Ethics:**

All procedures were reviewed and approved by IRB. I have no other concerns.

**Limitations:**

In Section 6, the authors mainly discuss the advantages of the proposed dataset and future works. I can not find the limitation discussion.

**Opportunities For Improvement:**

This work focuses on Isolated Sign Language Recognition (ISLR). As a non-expert, I still think that continuous sign recognition and sign language translation are more useful in real-world applications.
However, in Section 6, the authors clarify that the proposed dataset enables technologies like dictionary search, which are focused on classifying signs and do not require considering syntax or even optimal English translation.

My question is, whether the proposed dataset can benefit the CSLR or SLT tasks. The authors discourage researchers from using this
dataset alone (e.g., without also learning from continuous datasets) for more complex applications. But how about also using continuous datasets? Is the proposed dataset available and effective for CSLR or SLT, as the pre-training dataset or other basics? It is better to clarify or experimentally verify.

On the other hand, whether ISLR or say classifying signs, has some practical applications for the Deaf communities?

**Relation To Prior Work:**

This paper discussed how this work differs from previous works.

**Summary And Contributions:**

This work releases a  crowdsourced Isolated Sign Language Recognition, i.e., ASL Citizen, collected with consent and containing 83,399 videos for 2,731 distinct signs filmed by 52 signers in a variety of environments.

The authors propose that this dataset be used for sign language dictionary retrieval for American Sign Language (ASL), where a user demonstrates a sign to their webcam to retrieve matching signs from a dictionary.

Experimental results verify the effectiveness of this dataset that training supervised machine learning classifiers using it advances the state-of-the-art methods for dictionary retrieval task.

---

> ### Author Response · Authors · 2023-08-22
>
> Thank you for your thoughtful review of our work!
>
> We agree that continuous sign language translation would also be useful to many DHH users. Since translation is an unsolved problem, and since continuous sign language is much more than the sum of isolated signs (as present in our dataset), it is very difficult to speculate how our dataset might be used to further the translation task. Nonetheless, in order to learn grammar alongside vocabulary units, we can imagine ASL Citizen being combined with continuous sign language data and/or with a knowledge base of sign language grammar and linguistics. Recent advances in deep learning show promise for sign language modeling [3, 6], and it is possible that such pipelines could be engineered to incorporate single-sign examples as well as longer text, for example for pretraining.
>
> The dictionary task we outline in the paper also has its own merits. A dictionary is a fundamental resource for any language. Sign language users need functional dictionaries, just like anyone else. No functional sign language dictionaries that enable looking up a sign by demonstrating it currently exist. Since sign languages are their own distinct languages, this is functionally equivalent to e.g. all Spanish dictionaries requiring users to look up entries using English and thus operate in a non-native language. Sign language dictionaries are an established problem in HCI literature (e.g., [1, 5]) , and there are many successful commercial English-to-ASL dictionaries [7, 8, 9] but none that are ASL-to-English or ASL-to-ASL. In addition, sign language dictionaries further the documentation of sign languages, which are of great significance to Deaf culture. Beyond dictionaries, isolated sign recognition can enable creation of ASL-first user interfaces, for example to create a digital personal assistant that can respond to signed commands or short responses (e.g., [2, 3]).
>
> We discussed limitations through the discussion section, but will revise the draft to separate out the limitations section.
>
> [1] Danielle Bragg, Kyle Rector, and Richard E. Ladner. 2015. A User-Powered American Sign Language Dictionary. In Proceedings of the 18th ACM Conference on Computer Supported Cooperative Work & Social Computing (CSCW '15). Association for Computing Machinery, New York, NY, USA, 1837–1848. https://doi.org/10.1145/2675133.2675226
>
> [2] Abraham Glasser, Matthew Watkins, Kira Hart, Sooyeon Lee, and Matt Huenerfauth. 2022. Analyzing Deaf and Hard-of-Hearing Users’ Behavior, Usage, and Interaction with a Personal Assistant Device that Understands Sign-Language Input. In Proceedings of the 2022 CHI Conference on Human Factors in Computing Systems. 1–12.
>
> [3] Jens Forster, Christoph Schmidt, Oscar Koller, Martin Bellgardt, and Hermann Ney. 2014. Extensions of the Sign Language Recognition and Translation Corpus RWTH-PHOENIX-Weather. In International Conference on Language Resources and Evaluation. Reykjavik, Island, 1911–1916.
>
> [4] Abraham Glasser, Vaishnavi Mande, and Matt Huenerfauth. 2021. Understanding deaf and hard-of-hearing users' interest in sign-language interaction with personal-assistant devices. In Proceedings of the 18th International Web for All Conference (W4A '21). Association for Computing Machinery, New York, NY, USA, Article 24, 1–11. https://doi.org/10.1145/3430263.3452428
>
> [5] Saad Hassan, Akhter Al Amin, Alexis Gordon, Sooyeon Lee, and Matt Huenerfauth. 2022. Design and Evaluation of Hybrid Search for American Sign Language to English Dictionaries: Making the Most of Imperfect Sign Recognition. In Proceedings of the 2022 CHI Conference on Human Factors in Computing Systems (CHI '22). Association for Computing Machinery, New York, NY, USA, Article 195, 1–13. https://doi.org/10.1145/3491102.3501986
>
> [6] Oscar Koller, Sepehr Zargaran, Hermann Ney, and Richard Bowden. 2018. Deep Sign: Enabling Robust Statistical Continuous Sign Language Recognition via Hybrid CNN-HMMs. International Journal of Computer Vision 126, 12 (Dec. 2018), 1311–1325. DOI: http://dx.doi.org/10.1007/s11263-018-1121-3
>
> [7] Lifeprint. ASL Dictionary, 2023. URL,  https://www.lifeprint.com/dictionary.htm
>
> [8] Sign ASL. American Sign Language Dictionary, 2023. URL, https://www.signasl.org/
>
> [9] Signing Savvy. Signing savvy an ASL sign language video dictionary, 2022. URL https://www. signingsavvy.com/.

---

### Official Review · Reviewer_j3yN · 2023-07-20
**The paper is about  ASL (American Sign Langauge), crowdsourced Isolated Sign Language Recognition (ISLR) dataset.**

**Rating:** 6
**Confidence:** 4
**Correctness:** 1. The dataset is created nicely. It …
**Clarity:** Yes, the paper is well written.

**Strengths:**

1. The dataset being proposed is four times larger in scale compared to the currently available datasets in the Isolated Sign Language Recognition problem.
2. The ASL Citizen dataset tackles the problem of Isolated Sign Language Recognition, which is the utmost need in society's well-being for interaction with deaf people through sign language.
3. The authors improve state-of-the-art ISLR accuracy by more than double.
4.  The authors provide metrics for the dictionary retrieval task:  Mean Reciprocal Rank and Discounted Cumulative Gain.

**Additional Feedback:**

1. Perform baselining on more models and similar datasets for comparison.
2. Show GradCam or other visual explainable method results on your performed baseline models to strengthen your reported results. Since face and hand gestures are being used for the ISR task, hence to determine whether the existing models focus (on your dataset) on just face or hand or capture both aspects.

**Documentation:**

Yes, the paper presents sufficient detail on data collection and organization, availability and maintenance, and ethical and responsible use.
Benchmarking is not sufficient for comparison but codes for mentioned benchmarking models in the paper are available for reproducibility.

**Ethics:**

No ethical concerns.

**Limitations:**

1. There are very few baseline results. Just two models are shown in the paper for baselining.
2. This work primarily focuses on Sign Language used in North America, it lacks diversity in race.

**Opportunities For Improvement:**

1. Perform baselining on more models and similar datasets for comparison.
2. Show GradCam or other visual explainable method results on your performed baseline models to strengthen your reported results. Since face and hand gestures are being used for the ISR task, hence to determine whether the existing models focus (on your dataset) on just face or hand or capture both aspects.

**Relation To Prior Work:**

The paper shows comparison to only one existing dataset and two models being used for baselining.

**Summary And Contributions:**

1. The paper is about  ASL (American Sign Langauge), crowdsourced Isolated Sign Language Recognition (ISLR) dataset. The paper represents the importance of sign language for communication among deaf people.
2.  The dataset is collected for sign language translation. The authors provide a benchmark dataset and metrics for the dictionary retrieval task in a real-world setting.
3. The proposed dataset is built on the existing "Exploring 395 collection of sign language videos through crowdsourcing" dataset.
4. The dataset contains isolated sign videos from diverse signers in real-world settings representative of real-world dictionary queries.
5. It is the largest dataset among existing isolated sign datasets.
6. The authors performed baselining on two algorithms:  I3D and ST-GCN.

---

> ### Author Response · Authors · 2023-08-22
>
> Thank you for your thoughtful review of our work! We focus on American Sign Language (ASL) in this work, which is the primary sign language used in North America. Even in North America within ASL, there is great racial diversity amongst users [1]. While there are different sign languages used across the world (such as British Sign Language, Indian Sign Language etc.), it would be inappropriate to group together under a single label or word all the different signs used internationally for that concept. Each sign language has a unique vocabulary, just as each spoken language has a distinct vocabulary.
>
> As comparison points, pre-existing ASL datasets are extremely limited. We chose WLASL as our primary comparison because it is closest to our dataset in terms of size and medium (RGB videos) and because it is the current benchmark dataset for ISLR [2]. Similarly, our literature review of ISLR methods yielded two main types of approaches: appearance-based and pose-based techniques. As our focus was the dataset, we aimed to offer simple baselines that showcase the types of ISLR models in the literature, not state-of-the-art models. We will update the text in our abstract and introduction to reflect that our goal was to offer simple baselines. By choosing simple representative models, I3D and ST-GCN, we aimed to showcase the quality of the data. More complicated pipelines (such as those containing extensive pre- and post-processing with multiple training modules) might perform better, but may do so by encoding more assumptions about the data into methods and thus obfuscate the impact of the dataset. The simplicity of the chosen baseline architectures combined with the large improvements in performance are evidence of the quality of the data we have collected. Future work includes building on our baselines and exploring different modeling techniques and their benefits.
>
> We thank you for your suggestion to use GradCam to visually explain our models, and determine which features are being learned. However, since these techniques are often used on image models, it is unclear how they may generalize to a case where multiple frames are present as in our video scenario. For example, it is unclear if saliency on the hands is required in every frame, or if models can still extract relevant information by distributed saliency across frames or by focusing on the differences between frames, which is more difficult to verify by human impressions of GradCam maps. Explainability is an important avenue of inquiry, and makes for exciting future work, in particular for researchers focused on improving sign language modeling techniques.
>
> [1] Carolyn McCaskill,Ceil Lucas, Robert Bayley,and Joseph Christopher Hill. The hidden treasure of Black ASL: Its history and structure. Gallaudet University Press Washington, DC, 2011.
>
> [2] Dongxu Li, Cristion R. Opazo, Xin Yu and Hongdong Li, "Word-level Deep Sign Language Recognition from Video: A New Large-scale Dataset and Methods Comparison," 2020 IEEE Winter Conference on Applications of Computer Vision (WACV), Snowmass, CO, USA, 2020, pp. 1448-1458

---

> > ### Comment · Reviewer_j3yN · 2023-08-25
> > **Reply to addressed comments.**
> >
> > Thank you for addressing the comments.
> >
> > Dataset paper requires sufficient baselining results to showcase your data's importance experimentally over existing datasets and how existing algorithms are performing on your dataset.
> > This line "Future work includes building on our baselines and exploring different modeling techniques and their benefits." should not be the future work, it should be done in the dataset paper only as baselining results.

---

> > > ### Author Response · Authors · 2023-08-30
> > >
> > > Our primary contribution is the dataset, which is new and improved over prior datasets in several important ways -- larger in size, more representative of real-world users and environments, and collected with compensation and consent. It is also novel in that it is the first publicly available crowdsourced sign language dataset. Our baselines are emblematic of the primary modeling approaches to date, and so do demonstrate the value of our data. Lack of data is the primary barrier to sign language modeling, and we provide a valuable resource to help with this problem facing the entire field.

---

> > > > ### Comment · Reviewer_j3yN · 2023-08-30
> > > > **Dataset importance.**
> > > >
> > > > Increasing the rating since dataset is good and helpful to the community.

---

### Official Review · Reviewer_BfhD · 2023-07-22
**ASL Citizen: A Community-Sourced Dataset for Advancing Isolated Sign Language Recognition**

**Rating:** 8
**Confidence:** 5
**Clarity:** It is well written.

**Strengths:**

The new dataset, ASL Citizen, is now the largest ASL data set for ISLR. Some of the innovations include data collection through crowdsourcing; involvement of the deaf community throughout the data collection; all the subjects have given their consent in the use of sign data for research; transparency in the procedure; the recordings are done by the subjects themselves in their own environments resulting in-the-wild data. As a result, the authors claimed that the good ISLR performance is due to (1) the amount of data and (2)  the ample variations in the data, even though the test signs are recorded by signers unseen in the training data. The impact of the amount of data on the ISLR performance is also studied.

A major problem with crowdsourced data is that sometimes their quality cannot be guaranteed and there are a lot of noises. It is good to see that the authors have taken necessary steps to check and clean the data, and the use of "seed signs" is a good idea.

**Additional Feedback:**

N/A

**Correctness:**

I have some concern about the poor  ISLR result of WLASL-2000 which is an 2017 result and is about 40% worse than the SOTA result on the dataset.

**Documentation:**

very good.

**Ethics:**

no problem. It has passed IRB across their institutions.

**Limitations:**

The new dataset is aimed for ISLR but the more important tasks are CSLR and CSLT. (Just compare the difference between isolated word recognition vs. continuous speech recognition is automatic speech recognition.) So an ISLR dataset is not as useful as a CSLR/SLT dataset. Having said that an ISLR dataset may be used to improve the training of CSLR/SLRT to a certain extent.

**Opportunities For Improvement:**

The authors try to find an application for their ISLR effort, namely dictionary retrieval using sign queries. I am not convinced that that is an important task. On the other hand, sign spotting from a long sign video (with a sign word) may be more useful. In any case, the ISLR dataset can still be useful for ISLR or for training the model for continuous SLR (e.g., initialization, pre-training, etc.)

A major concern to me is that the quoted poor performance of WLASL-2000 is from an old 2017 paper using I3D. Since then there are many improvements and the latest result I know is 61.26% (from Ronglai Zuo, Fangyun Wei, Brian Mak, "Natural Language-Assisted Sign Language Recognition," CVPR 2023. Table 2 in the paper also gives performances from other models including HMA, TCK, SignBERT, SAM, etc. all with accuracies in the 50+%.)  And the test signers in WLASR-2000 are also unseen. Although the authors also presented the comparison on the overlapping subset, it is not clear if the better performance is due to their better data or poor model.

**Relation To Prior Work:**

Good citations.

**Summary And Contributions:**

The paper is a major extension of the authors' effort in collecting ASL dataset using crowdsourcing from their pilot study in [10]. It is a major effort in using crowdsourcing for collecting ASL data for isolated SLR (ISLR) or more particularly for dictionary retrieval using sign query. The paper emphasizes that the collection involves deaf researchers and the deaf community with great transparency and consent from the contributors, and all procedures have been reviewed and approved by IRB by the involving institutions. At the end, the new dataset is the largest ASL dataset for ISLR in terms of the vocabulary size and no. of videos per word, and the videos are more of the "in-the-wild" style with diverse visual backgrounds and the signers' age ranges from 20-72 years coming from 16 US states. Benchmark results provided by recent ISLR models are given are compared with those from another major ASL ISLR dataset, the WLASL-2000 and the improvement is about double with an accuracy of 63%.

---

> ### Author Response · Authors · 2023-08-22
>
> Thank you for your thoughtful review of our work!
>
> We agree that continuous sign language translation would also be useful to many DHH users. However, the dictionary task we outline in the paper has its own merits. A dictionary is a fundamental resource for any language. Sign language users need functional dictionaries, just like anyone else. No functional sign language dictionaries that enable looking up a sign by demonstrating it currently exist. Since sign languages are their own distinct languages, this is functionally equivalent to e.g. all Spanish dictionaries requiring users to look up entries using English and thus operate in a non-native language. Sign language dictionaries are an established problem in HCI literature (e.g., [1, 4]) , and there are many successful commercial English-to-ASL dictionaries [5, 6, 7] but not vice versa. In addition, sign language dictionaries further the documentation of sign languages, which are of great significance to Deaf culture. Beyond dictionaries, isolated sign recognition can enable creation of ASL-first user interfaces, for example to create a digital personal assistant that can respond to signed commands or short responses (e.g., [2, 3]). We believe this dataset could also be beneficial in the sign spotting task you suggest, but alongside a closely related continuous sign language dataset.
>
> We thank you for noting the CVPR 2023 paper. This paper had not been released at the time of our submission, and we will discuss it in our related work.
>
> Because our primary contribution is our dataset, we did not seek state-of-the-art models as baselines. We will update the text in our abstract and introduction to reflect that our goal was to offer simple baselines. By choosing simple models like I3D and ST-GCN, we aimed to showcase the quality of the data. A more complicated pipeline (such as those containing extensive pre- and post-processing with multiple training modules) might perform better with a given dataset, but may do so by encoding more assumptions about the data into methods. For example, a pose-based model relies on accurate extraction of keypoints, but it is known that obfuscations negatively impact accuracy of keypoints. Similarly, gloss-smoothing as suggested in [CVPR paper] relies on semantic relations between glosses that are hard to acquire and may not hold in the future. These assumptions and resulting complexity may then obscure the impact of the dataset. Models with fewer assumptions (and simpler architectures) additionally have the benefits of being more generalizable, more accessible, and perhaps better suited to certain contexts (e.g., mobile applications with limited computational resources). Since WLASL does not guarantee unseen user splits by virtue of the random assignment used, the ASL Citizen test split proposes a more difficult task. The jump in performance then between I3D models trained on ASL Citizen vs. WLASL is indicative of the quality of data in ASL Citizen. We leave it to future work to build on our baselines and explore different modeling techniques and their benefits.
>
> [1] Danielle Bragg, Kyle Rector, and Richard E. Ladner. 2015. A User-Powered American Sign Language Dictionary. In Proceedings of the 18th ACM Conference on Computer Supported Cooperative Work & Social Computing (CSCW '15). Association for Computing Machinery, New York, NY, USA, 1837–1848. https://doi.org/10.1145/2675133.2675226
>
> [2] Abraham Glasser, Matthew Watkins, Kira Hart, Sooyeon Lee, and Matt Huenerfauth. 2022. Analyzing Deaf and Hard-of-Hearing Users’ Behavior, Usage, and Interaction with a Personal Assistant Device that Understands Sign-Language Input. In Proceedings of the 2022 CHI Conference on Human Factors in Computing Systems. 1–12.
>
> [3] Abraham Glasser, Vaishnavi Mande, and Matt Huenerfauth. 2021. Understanding deaf and hard-of-hearing users' interest in sign-language interaction with personal-assistant devices. In Proceedings of the 18th International Web for All Conference (W4A '21). Association for Computing Machinery, New York, NY, USA, Article 24, 1–11. https://doi.org/10.1145/3430263.3452428
>
> [4] Saad Hassan, Akhter Al Amin, Alexis Gordon, Sooyeon Lee, and Matt Huenerfauth. 2022. Design and Evaluation of Hybrid Search for American Sign Language to English Dictionaries: Making the Most of Imperfect Sign Recognition. In Proceedings of the 2022 CHI Conference on Human Factors in Computing Systems (CHI '22). Association for Computing Machinery, New York, NY, USA, Article 195, 1–13. https://doi.org/10.1145/3491102.3501986
>
> [5] Lifeprint. ASL Dictionary, 2023. URL,  https://www.lifeprint.com/dictionary.htm
>
> [6] Sign ASL. American Sign Language Dictionary, 2023. URL, https://www.signasl.org/
>
> [7] Signing Savvy. Signing savvy an ASL sign language video dictionary, 2022. URL https://www. signingsavvy.com/.

---

> > ### Comment · Reviewer_BfhD · 2023-08-26
> >
> > I appreciate the authors' answers.
> >
> > Personally, I am just curious how often the deaf user will use a sign dictionary as said.
> >
> > For using simple or complex models: I agree that you should use models that generalize well, but that doesn't mean simple models. Anyway, yes, the CVPR2023 paper is published after NeurIPS submission, but I just want to refer you to Table2 of the paper which lists out a bunch of WLASL-2000 ISR results which are all over 50%.
> >
> > In any case, I still support the acceptance of this paper.

---

> > > ### Author Response · Authors · 2023-08-30
> > >
> > > Thank you for the clarification! In light of these other works as well as the CVPR paper, we have revised claims about our classifiers trained on ASL-Citizen to specify that they achieve competitive performance rather than advancing the state-of-the-art absolute reported accuracy. We also revised our related work section to discuss these advances in ISLR.

---

### Official Review · Reviewer_UkwP · 2023-07-23
**Useful, sensitive, and well justified dataset**

**Rating:** 9
**Confidence:** 4
**Correctness:** The methods and results look robust.
**Clarity:** The paper is clearly written.

**Strengths:**

The models built using this dataset make a very impressive leap in performance, and do so while maintaining high standards in terms of data quality and ethics.

**Additional Feedback:**

Out of curiosity I'd be interested to know more about how the work could be extended toward making a translation system, and also how the Deal and HoH communities feel about this possibility. Is that a desirable product?

**Documentation:**

Intended uses are well covered, as are the ethical concerns.

**Ethics:**

Ethics was well covered in the paper. I'm satisfied that they used appropriate consent procedures, protected privacy, and showed concern for responsible use.

**Limitations:**

The authors dealt thoroughly with potential objections from Deaf and HoH communities, which were the most obvious concern.

**Opportunities For Improvement:**

As the discussion mentions up front, this dataset is not suitable for ASL translation purposes. It makes significant strides in a direction that might eventually lead there though.

**Relation To Prior Work:**

I don't know the literature, so can't comment with authority, but there is a section covering previous literature that has the appearance of being thorough.

**Summary And Contributions:**


The contribution is an ASL dataset, consisting of videos of signs by many Deaf or HoH signers for use in dictionary lookup applications.

---

> ### Author Response · Authors · 2023-08-22
>
> Thank you for your thoughtful review of our work!
>
> The continuous sign language translation task you suggest does have value to the community since it has the potential to address the gap in access to information and communication. Mainstream communication (emails, newspapers, TV, in-person discussions, etc.) typically occur in spoken or written language. Because this language is not accessible to many DHH people, professional sign language interpreters are often hired to interpret or translate. However, human interpreters are not always available [4] and can be prohibitively expensive, leaving the opportunity for automatic translation to fill the gap. Prior works have noted both the utility and considerations that should go into designing these translation systems [1, 2, 6].
>
> Since translation is an unsolved problem, and since continuous sign language is much more than the sum of isolated signs (as present in our dataset), it is very difficult to speculate how our dataset might be used to further the translation task. Nonetheless, in order to learn grammar alongside vocabulary units, we can imagine ASL Citizen being combined with continuous sign language data and/or with a knowledge base of sign language grammar and linguistics. Recent advances in deep learning show promise for sign language modeling [3, 5], and it is possible that such pipelines could be engineered to incorporate single-sign examples as well as longer text, for example for pretraining.
>
> [1] Danielle Bragg, Oscar Koller, Mary Bellard, Larwan Berke, Patrick Boudreault, Annelies Braffort, Naomi Caselli, Matt Huenerfauth, Hernisa Kacorri, Tessa Verhoef, Christian Vogler, and Meredith Ringel Morris. 2019. Sign Language Recognition, Generation, and Translation: An Interdisciplinary Perspective. In The 21st International ACM SIGACCESS Conference on Computers and Accessibility (Pittsburgh, PA, USA) (ASSETS ’19). Association for Computing Machinery, New York, NY, USA, 16–31. https://doi.org/10.1145/3308561.3353774
>
> [2] Maartje De Meulder. 2021. Is “good enough” good enough? Ethical and responsible development of sign language technologies. In Proceedings of the 1st International Workshop on Automatic Translation for Signed and Spoken Languages (AT4SSL), pages 12–22, Virtual. Association for Machine Translation in the Americas.
>
> [3] Jens Forster, Christoph Schmidt, Oscar Koller, Martin Bellgardt, and Hermann Ney. 2014. Extensions of the Sign Language Recognition and Translation Corpus RWTH-PHOENIX-Weather. In International Conference on Language Resources and Evaluation. Reykjavik, Island, 1911–1916.
>
> [4] Nimisha Jaiswal. 2017. With a deaf community of millions, hearing India is only just beginning to sign.(2017).https://theworld.org/stories/2017-01-04/deaf-community-millions-hearing-india-only-just-beginning-sign
>
> [5] Oscar Koller, Sepehr Zargaran, Hermann Ney, and Richard Bowden. 2018. Deep Sign: Enabling Robust Statistical Continuous Sign Language Recognition via Hybrid CNN-HMMs. International Journal of Computer Vision 126, 12 (Dec. 2018), 1311–1325. DOI: http://dx.doi.org/10.1007/s11263-018-1121-3
>
> [6] World Federation of the Deaf. [n.d.]. WFD and WASLI Statement on Use of Signing Avatars. https://wfdeaf.org/news/resources/wfd-wasli-statement-use-signing-avatars/ Accessed 2023-03-31.

---

### Author Response · Authors · 2023-08-30
**Summary of revisions**

We thank all reviewers for their time and feedback! Your feedback has helped us improve our paper, and we have uploaded a revised version of our paper with edits highlighted. Our main changes are listed below:
- We updated our abstract, introduction and methods description to reflect our aim is to offer generalizable baselines for the community to build upon.
- We edited our related work section to include recent advances in ISLR that achieve better performance, and discuss our rationale for our chosen baseline architectures.
- We toned down claims about our general classifiers trained on ASL Citizen to specify that they achieve competitive performance rather than significantly advancing the state-of-the-art accuracy.
- We emphasize the value of dictionaries for signing communities, and discuss other applications for ISLR like ASL-first user interfaces through the paper.
- Lastly, we separate out our limitations in a discussion subsection.

---

### Decision · Program_Chairs · 2023-09-22

**Decision:**

Accept (Poster)

**Comment:**

The paper delivers the first crowdsourced Isolated Sign Language Recognition (ISLR) dataset, termed ASL Citizen. As pointed by Reviewer BfhD, this is "the largest ASL dataset for ISLR in terms of the vocabulary size and no. of videos per word, and the videos are more of the "in-the-wild" style with diverse visual backgrounds and the signers' age ranges from 20-72 years coming from 16 US states".  Authors propose to use the dataset for sign language dictionary retrieval task, where a user demonstrates a sign to their webcam to retrieve matching signs from a dictionary.

All Reviewers are positive about the dataset and acknowledge
- the need for such dataset, e.g. pointed by Reviewer j3yN "which is the utmost need in society's well-being for interaction with deaf people through sign language"
- (Reviewer UkwP) well covered ethics with high standards e.g. "the authors dealt thoroughly with potential objections from Deaf and HoH communities, which were the most obvious concern.", including involvement of deaf researcher and community into data construction (pointed by Reviewer BfhD)
- (Reviewer BfhD) high quality of data: "clean the data, and the use of "seed signs" is a good idea."

The main two concerns raised by Reviewers are
- (by Reviewer BfhD) The issues with benchmarking baseline models on WLASL-2000 vs ASL Citizen: models trained on WLASL-2000 have significantly worse performance. Though authors tried to address this concern, I still agree with the Reviewer BfhD, and believe more advanced models should be benchmarked also as then we can see the trend how larger/improved dataset collection affects the latest state-of-the-art models and how both of them could benefit from each other. Also this could confirm the necessity of better data collection as it can improve current models.
- (by Reviewers UkwP, BfhD, RCuQ) Continuous sign recognition and sign language translation are more useful in real-world applications, and proposed isolated sign recognition is a very limited task, e.g. not clear how dictionary retrieval using sign queries is an important task. I agree with the position of Reviewers, that continuous sign language recognition is way harder, more interesting task with more obvious applications (compare speech recognition to audio classification, or commands recognition). At the same time I believe collecting ASL Citizen could be the push to the community to further develop datasets for continuous sign recognition as we also started from speech commands and simpler data in the speech recognition domain.

Having high quality data and high standards of ethics, the importance of the dataset to the community, I support acceptance of the paper despite the above 2 main concerns.